# The transcription factor STAT5 catalyzes Mannich ligation reactions yielding inhibitors of leukemic cell proliferation

Ee Lin Wong[1], Eric Nawrotzky[1], Christoph Arkona[1], Boo Geun Kim[2], Samuel Beligny[2], Xinning Wang[1], Stefan Wagner[1], Michael Lisurek[2], Dirk Carstanjen[2] & Jörg Rademann [1,2]

Protein-templated fragment ligations have been established as a powerful method for the assembly and detection of optimized protein ligands. Initially developed for reversible ligations, the method has been expanded to irreversible reactions enabling the formation of super-additive fragment combinations. Here, protein-induced Mannich ligations are discovered as a biocatalytic reaction furnishing inhibitors of the transcription factor STAT5. STAT5 protein catalyzes multicomponent reactions of a phosphate mimetic, formaldehyde, and 1$H$-tetrazoles yielding protein ligands with greatly increased binding affinity and ligand efficiency. Reactions are induced under physiological conditions selectively by native STAT5 but not by other proteins. Formation of ligation products and (auto-)inhibition of the reaction are quantified and the mechanism is investigated. Inhibitors assembled by STAT5 block specifically the phosphorylation of this protein in a cellular model of acute myeloid leukemia (AML), DNA-binding of STAT5 dimers, expression of downstream targets of the transcription factor, and the proliferation of cancer cells in mice.

---

[1] Department of Biology, Chemistry and Pharmacy, Medicinal Chemistry, Freie Universität Berlin, Königin-Luise-Str. 2+4, 14195 Berlin, Germany.
[2] Department of Medicinal Chemistry, Leibniz Institut für Molekulare Pharmakologie (FMP), Robert-Rössle-Str. 10, 13125 Berlin, Germany. Correspondence and requests for materials should be addressed to J.R. (email: joerg.rademann@fu-berlin.de)

In protein-templated fragment ligations, proteins catalyze the formation of ligands with enhanced binding affinity from bound reactive fragments, thereby integrating chemical synthesis of protein ligands and modulation of protein activity in one single step[1–6]. The chemical space accessible by protein-templated reactions, however, is still strongly limited. Whereas most applications so far have employed reversible[7–10], i.e., dynamic ligation reactions, few reactions have been used for the irreversible[11–15] formation of stable protein ligands by protein-templated reactions, mostly employing dipolar cycloaddition reactions[13,14]. Recently, amide bonds, a privileged fragment linkage in bioactive molecules, have been accessed by protein-templated chemistry[16]. Evidently, it would be a major expansion of the chemical space populated by protein-dependent reactions, if multicomponent reactions could be employed for ligand formation—possibly exploiting alternative mechanisms[9].

Here, Mannich ligations, three-component reactions of an amine, aldehyde, and N- or C-nucleophile, are investigated as protein-catalyzed reactions for discovering protein ligands. As a target, the transcription factor STAT5 (STAT = signal transducers and activators of transcription) is selected representing a central hub in the signaling of numerous cancer cell types. A small heterocyclic phosphate-mimetic binding to STAT5 with low affinity is discovered from a fragment screen and used as a starting point for fragment expansion using protein-induced multicomponent reactions. Mannich ligation products with improved affinity for STAT5 are assembled by the protein and selectivity, pH-dependence, and the biocatalytic mechanism of this protein-induced reaction are investigated. Furthermore, selectivity and functional efficacy of the inhibitors are validated for isolated proteins, in complex cell lysates, in living cells and finally in mice in order to evaluate the functional potential of protein ligands generated by protein-induced ligation.

## Results

**Target selection**. The transcription factor STAT5 was selected as a target owing to its central role in the signaling of numerous cancer cell types[17,18]. STAT5 is expressed in two closely related forms, STAT5a and b, with 94% identity of the amino-acid sequence. STAT5 contains an SH2 domain, which is responsible for the binding of the protein to phosphotyrosine-containing peptide sequences of kinases, resulting in the phosphorylation at tyrosine 694/699. Phospho-STAT5 dimerizes via its SH2 domain and translocates into the nucleus[19]. There, p-STAT5 dimers bind to genomic promoter sequences inducing the transcription of target genes. In consequence, ligands of STAT-SH2 domains are able to suppress STAT signaling and have been recognized as potential anticancer drugs. STATs, however, have been described as difficult pharmacological targets[20]. Although highly potent peptide ligands have been reported for the STAT-SH2 domains[21,22], these phosphopeptide mimetics usually fail to provide significant cellular activity. Several cellularly active small molecule inhibitors of STAT5 have been developed as well, however, none of these have been based on protein-induced reactions yet[23–25].

**Discovery and validation of a phosphate-mimetic fragment**. Binding of fragments to the recombinantly expressed STAT5b-SH2 domain fused to maltose-binding protein (MBP) as affinity tag was recorded by measuring fluorescence polarization (FP) of the carboxyfluoresceine-labeled phosphotyrosine octapeptide **1** ($K_D = 55$ nM) (Fig. 1a)[21]. A collection of 17,000 fragments and fragment combinations composed in accordance with the substructure composition of the World Drug Index (WDI)[26] was screened for inhibiting the phosphopeptide-STAT5b interaction. Primary amine fragments were tested in the FP assay in the presence of electrophilic phosphotyrosine mimic **2** as described earlier for protein tyrosine phosphatases (PTP) in order to distinguish secondary site binders that enhance the inhibition of **2** from inhibitors that are not affected by **2**[5,21]. From the second group, one fragment, 4-amino-furazan-3-carboxylic acid **3** (M = 129 g mol$^{-1}$, Fig. 1a) displayed a $K_D$ value of 420 μM, corresponding to the ligand efficiency of 2.1 kJ mol$^{-1}$ per non-hydrogen atom, higher than that of the nanomolar phospho-peptide **1**, the phosphotyrosine-mimetic **2**, and the best reported STAT5 inhibitors[23–25]. Ligands with such high ligand efficiency are rather found for enzymatic binding pockets than for protein–protein interaction sites and thus fragment **3** was selected for further validation[27]. Binding of **3** to STAT5b-SH2 was confirmed using the thermofluor assay[28,29], a thermal shift assay (TSA), as an independent biophysical assay. Binding of fragment **3** augmented the melting point of STAT5 by $\Delta T_m$ of 3 °C (Supplementary Figure 1). Potential binding modes of the phosphotyrosine **2** and the fragment hit **3** were scrutinized using a homology model of STAT5b derived from the crystal structure of STAT5a (PDB:1Y1U [https://doi.org/10.2210/pdb1Y1U/pdb]) for molecular docking (Fig. 1b, c)[30]. The phosphotyrosine binding site in the STAT5-SH2 domain is shallow compared with the deeper binding pockets of PTP[31,32], coordinating phenyl phosphate **2** by only two amino-acid residues, Arg618 and Ser622. As

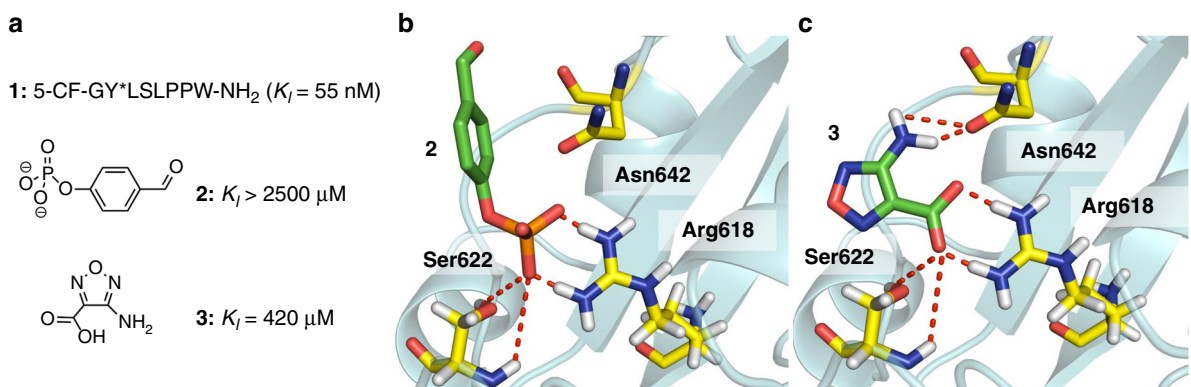

**Fig. 1** Discovery of phosphate-mimetic fragment **3**. **a** Fluorescently labeled phosphotyrosine peptide **1** was used in an FP assay for the screening of a fragment library furnishing 4-amino-furazan-3-carboxylic acid **3** as a phosphate-mimetic[21]. Phosphotyrosine-mimetic fragment 4-formyl-phenyl phosphate **2** was employed to investigate fragment hits for second site binding. **b–c** Molecular docking results of fragments **2** and **3** into homology model of human STAT5b-SH2 domain, generated from the published structure of STAT5a (PDB accession codes, 1Y1U [http://dx.doi.org/10.2210/pdb1Y1U/pdb])[30]. Hydrogen bonds with key residues in the hydrophilic binding pocket of the STAT5-SH2 domain were illustrated as red dashed lines

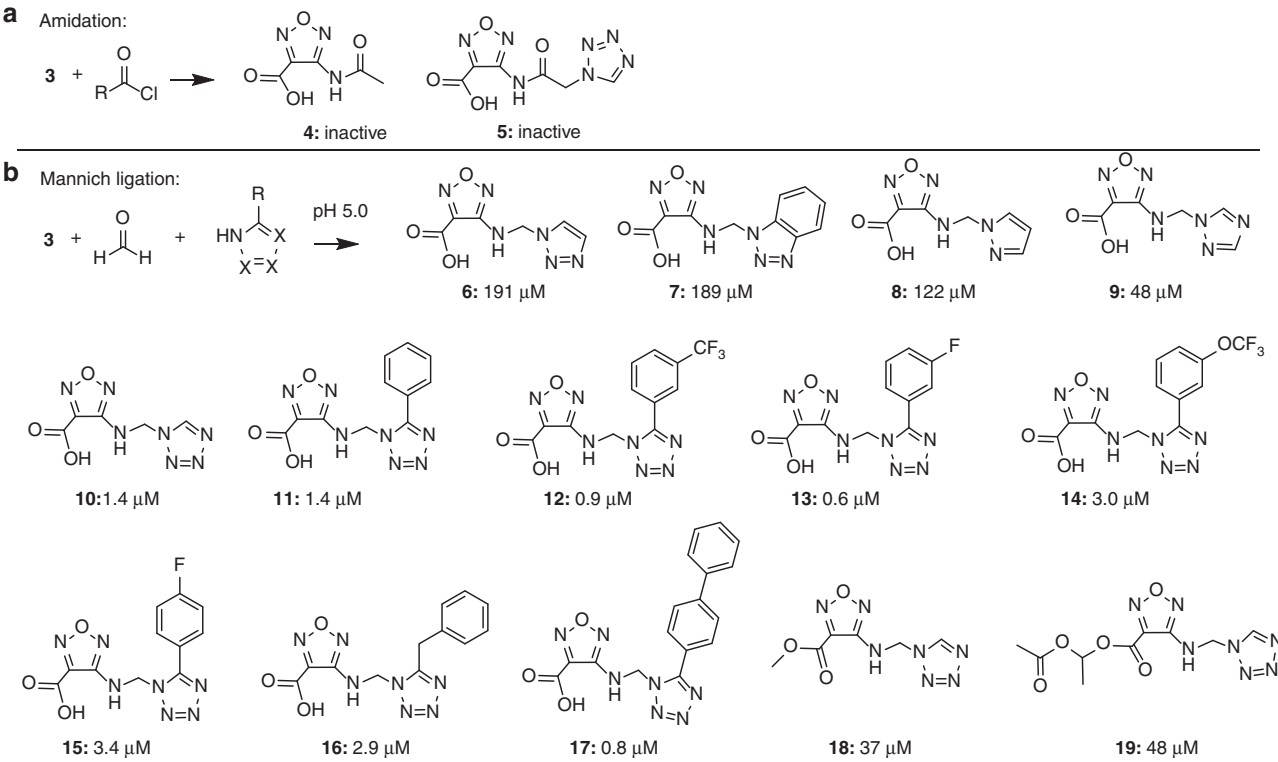

**Fig. 2** Expansion of fragment **3** through protein-induced reactions. **a** Amidation of **3** yielded compounds **4** and **5**, which were inactive in the FP assay. **b** Mannich ligation was investigated as an alternative fragment expansion method to obtain the active compounds **6–19** containing a linker with reduced steric hindrance and better structural flexibility

a result, the benzene ring of **2** is not buried in a cavity like in the case of PTPs but rather exposed to the solvent at the protein surface. Binding of fragment **3** is mediated by the Coulomb interaction between the carboxylate anion and the cation of protonated Arg618 and H-bonds involving Arg618, Ser622, and Asn642.

**Fragment expansion via protein-induced Mannich ligations.** First, the discovered phosphate-mimetic **3** was expanded by amidation (Fig. 2a), a reaction recently introduced to protein-templated fragment ligations[16]. The N-acetyl derivative **4** and all other tested amides including **5**, however, were inactive in the FP assay. In order to reduce the steric demand from a carbonyl to the more flexible methylene linkage, the Mannich ligation was investigated as fragment expansion method (Fig. 2b). Fragment **3** was found to react readily with formaldehyde (FA) and various N-heterocycles in aqueous buffer at pH 5.0 at room temperature yielding Mannich ligation products, whereas no reaction was observed without protein at pH 7.4. In order to implement protein-dependent Mannich ligations, the compatibility of the reaction with the protein MBP-STAT5b-SH2 and with the FP assay was investigated. 3-(N-Morpholino)-propane sulfonic acid (MOPS, 50 mM, pH 7.4) was used as a buffer containing no primary and secondary amines that could interfere with the reaction. FP of MBP-STAT5b-SH2 (125 nM) with peptide **1** (10 nM) was recorded in the presence of increasing concentrations of FA at pH 7.4. No change of FP was observed at concentrations up to 250 μM FA, whereas at higher concentrations, FP values increased considerably (Fig. 3a). Likewise, up to 250 μM no effect of FA on the melting point of STAT5b-SH2 was recorded in the TSA, although higher FA concentrations reduced the intensity of the fluorescence signal, suggesting interference of FA with the

fluorescent dye (Fig. 3b). For in situ Mannich ligation assays, fragment **3** was incubated with one hetaryl nucleophile and FA (all 250 μM) per microtiter plate-well in water resulting in pH 5.0. After 12 h incubation at room temperature, protein MBP-STAT5b (125 nM in 50 mM MOPS buffer pH 7.4) with the peptide probe **1** were added and further incubated for 15 min before FP was recorded. Several of the added heterocycles led to substantially decreased FP values—suggesting the formation of a Mannich ligation product as inhibitor of STAT5b with increased affinity. Active Mannich ligation products were re-synthesized, purified, and tested in the FP assay (Fig. 2b, Supplementary Figures 11–13). Remarkably, the addition of five-membered N-heterocycles ("azoles") to the formimine of fragment **3** led to strongly enhanced inhibition of STAT5b. The 1,2,3-triazol-1-yl product **6** as well as the benzo-1,2,3-triazol-1-yl **7** increased the affinity by a factor of > 2, pyrazol-1-yl **8** > threefold, 1,2,4-triazol-1-yl **9** > ninefold and tetrazol-1-yl **10** 300-fold resulting in a $K_I$ of 1.4 μM (Supplementary Figure 2). The reaction with 5-substituted tetrazoles yielded strongly active inhibitors **11–17**, some even with submicromolar affinities, including 4-(5-phenyl-tetrazol-1-yl-methylamino)-furazane-3-carboxylate **11** (1.4 μM), 5-(3-tri-fluoromethyl-phenyl)- **12** (0.9 μM), 5-(3-fluorophenyl) **13** (0.6 μM), 5-benzyl **16** (2.9 μM), and 5-biphenyl **17** (0.8 μM). Esters of the furazane carboxylic acid (**18**, **19**) were prepared as prodrug derivatives. 4-(Tetrazolyl-1-methylamino)-furazan-3-carboxylic acid **10** is the STAT5 inhibitor with the highest ligand efficiency of 2.23 kJ mol$^{-1}$ per non-hydrogen atom. All starting azoles like tetrazole **25** were completely inactive at concentrations of 5 mM, thus the inhibitors constitute examples of super-additive fragment combinations. As a consequence, the observed protein-dependent ligation reaction did not proceed as a protein-templated reaction, that requires the binding of both reacting fragments to the protein.

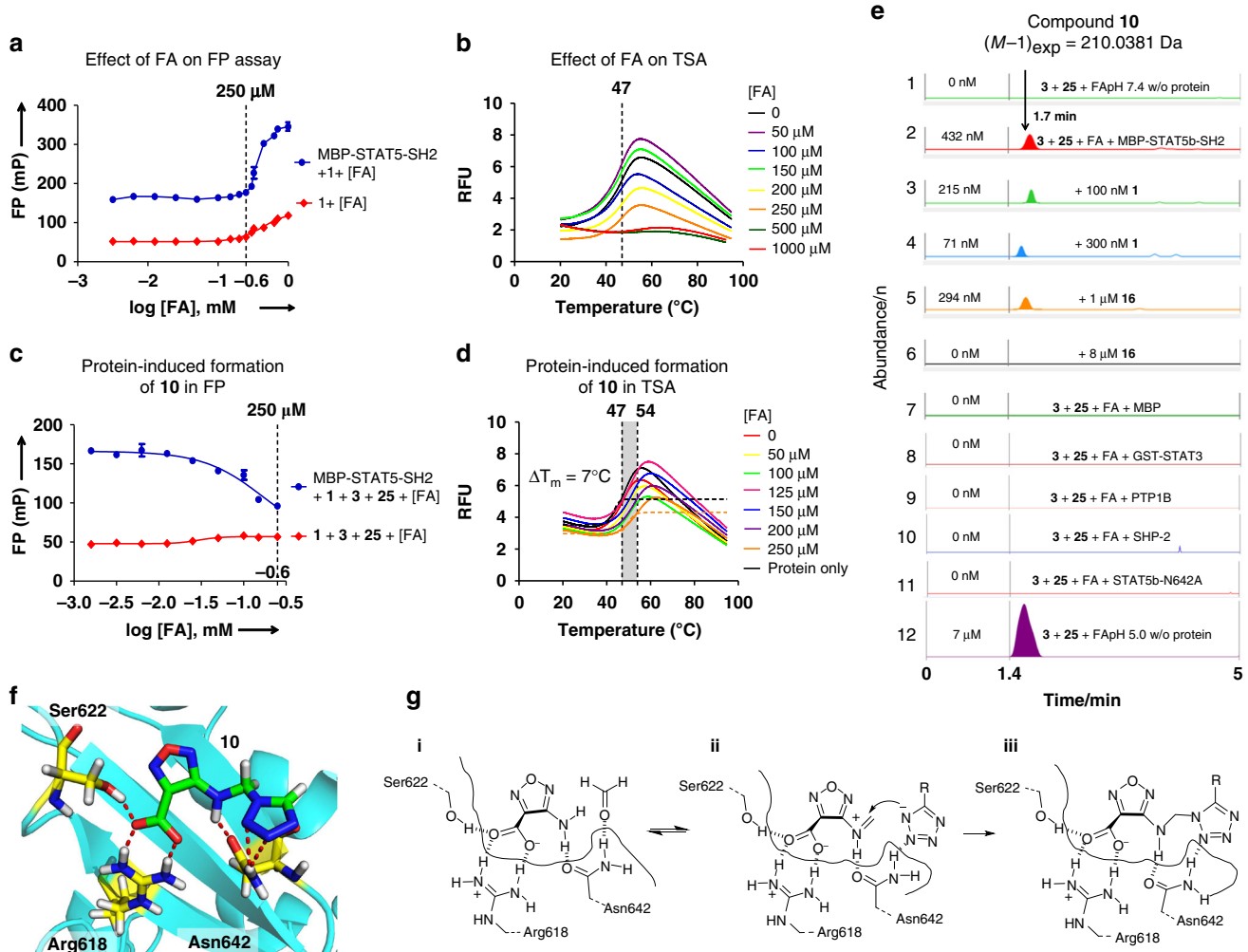

**Fig. 3** Assembly of STAT5 inhibitor **10** through protein-induced Mannich ligations. **a** FA was tolerated at up to 250 μM in the FP assay of MBP-STAT5b-SH2 ($n = 3$). **b** FA did not affect the STAT5b protein stability in the TSA at ≤ 250 μM (data shown are one representative of $n = 3$). **c** Protein-induced formation of **10** in the FP assay with increasing FA concentrations ($n = 3$). **d** Protein-induced formation of **10** from fragments **3** and $1H$-tetrazole **25** with increasing FA concentrations in the TSA ($\Delta T_m = 7 °C$) (data shown are one representative of $n = 3$). **e** Formation of **10** detected in the HPLC-QTOF-MS. 1: No formation of **10** from FA, fragments **3** and **25** (all 250 μM) at pH 7.4 in MOPS buffer without protein. 2: Protein-induced formation of compound **10** with protein MBP-STAT5b-SH2 (250 nM) at pH 7.4. 3–6: Inhibition of protein-induced formation of **10** by peptide **1** (3,4) or inhibitor **16** (5,6). 7,8: No formation of compound **10** in the presence of Maltose Binding Protein (1 μM) or GST-STAT3 (250 nM). 9–11: Compound **10** was not formed in the presence of phosphatases, SHP2 (250 nM) and PTP1B (250 nM) nor in the presence of mutant STAT5b-N642A at pH 7.4 in MOPS buffer. 12: Formation of compound **10** at pH 5.0 without protein. Data shown are one representative example of $n = 3$. **f** 3D-binding model of the STAT5b:**10** complex. Hydrogen bonds are illustrated as red dashed lines. **g**, Mechanism of the protein-induced formation of **10** from **3** with FA and **25** (R = H). (**i**) Binding of **3** via Arg618, Ser622, and Asn642, activation of FA with Asn642. (**ii**) Activation of the forminium cation of **3**, coordination of the incoming tetrazolium anion of **25** by Asn642 leads to formation of Mannich ligation product **10** (**iii**). Error bars denote mean ± S.D

**Mechanistic analysis of the protein-induced reactions**. Protein-catalyzed reactions of fragment **3** with FA and $1H$-tetrazole **25** were investigated by using the binding assays (FP, TSA), and high-performance liquid chromatography-mass spectrometry (HPLC-MS) analysis. Incubation of fragments **3** and **25** (250 μM each) with MBP-STAT5b-SH2 (250 nM) with increasing concentrations of FA at pH 7.4 led to decreased FP values, suggesting the formation of an inhibitor (Fig. 3c). The protein-induced formation of a STAT5 inhibitor was confirmed in the TSA: Incubation of fragments **3** and **25** with increasing concentrations of FA and the protein resulted in a shift of the protein's melting point $\Delta T_m$ by 7 °C (Fig. 3d). High-resolution HPLC-QTOF-MS analysis was employed to quantify Mannich ligation product **10** formed with or without protein present (Fig. 3e). At pH 7.4, absolutely no inhibitor was formed from **3**, **25**, and FA, if

MBP-STAT5-SH2 protein was not present (trace 1). With 250 nM MBP-STAT5-SH2 in the buffer at pH 7.4, 432 nM of **10** were formed over 24 h (average of three independent experiments). The protein-dependent reaction was saturated after 24 h, no significant changes in product concentration were observed between 24 and 48 h reaction time—suggesting product inhibition of the ligation reaction. Addition of phosphopeptide **1** or inhibitor **16** to the protein-induced reaction suppressed the formation of **10** completely or partly in a concentration-dependent manner (traces 3–6). If instead of the MBP-STAT5-SH2 protein only the protein tag MBP (1 μM) or the catalytic domains of tyrosine phosphatases PTP1B or SHP2 (250 nM) were added, no product was formed at all (traces 7,9,10). In contrast, incubation of reagents **3**, **25**, and FA at pH 5.0 led with or without protein to the formation of 7 μM of inhibitor **10** in a protein-independent

background reaction (trace 12). Similar data were obtained for the protein-dependent reaction of fragments **3**, FA, and benzyl-tetrazole **26** although traces of a background reaction were observed in this case (Supplementary Figure 3). In contrast, no protein-dependent reaction was observed when replacing the 1H-tetrazoles by 1,2,4-triazole (Supplementary Figure 3).

Molecular docking suggested that the large increase in binding affinity of compound **10** was contributed by expansion of the binding interaction into the adjacent amphiphilic pocket containing residues Trp641, Leu643, and Met639 via hydrophobic contacts. The H-bond, between the 4-amino group of **10** and the carbonyl of Asn642 was retained and possibly enforced by the higher polarity of the NH-bond in **10** (Fig. 3f). An additional H-bond was proposed by BINding ANALyzer (BINANA) between the tetrazole ring and the amide-NH$_2$ of Asn642 and strengthened the binding of compound **10** in the binding pocket[18,23,33,34]. In order to challenge the postulated importance of Asn642 for the reactivity of fragment **3** with STAT5b-SH2, the fragment ligation of **3**, FA, and 1H-tetrazole at pH 7.4 was investigated with GST-STAT3, a protein without Asn in the otherwise similar phosphotyrosine recognition site (Fig. 3, trace 8). With STAT3 instead of STAT5b absolutely no fragment ligation product was formed. In addition, the mutant MBP-STAT5b-SH2 N642A was generated by site-directed mutagenesis. The mutant protein displayed strongly reduced binding affinity to peptide **1** with $K_D$ of ca. 5 μM (instead of 55 nM for the wildtype) (Supplementary Figure 2g). Binding of the peptide was inhibited by compound **10** with an $IC_{50}$ of 30 μM (instead of 1.5 μM for the wildtype) (Supplementary Figure 2h). Accordingly, the conditions of the fragment ligation reaction yielded no product with the mutant protein as well. Plausibly, the side chain carbonyl group of Asn642 binds to the amino group of compound **3** via accepting an H-bond (Fig. 3g). As a result, the amide-NH$_2$ of Asn642 is free to coordinate and activate the incoming FA as an H-bond donor, leading to the formation of the formiminium derivative of **3**, which is again stabilized by its H-bond to the Asn-carbonyl. Next, the amide-NH$_2$ could coordinate the incoming 1H-tetrazolium ion which reacts to the observed ligation product **10**.

**Specificity of STAT5 inhibitors.** Compound **10** was tested with closely related STAT3 and the catalytic domain of protein tyrosine phosphatase SHP2 (PTPN10) and at concentrations up to 1 mM no binding or inhibition was observed (Supplementary Figure 4). Next, phosphorylated STAT-dimers were extracted from nuclei of BaF3/FLT3-ITD cells and binding of STAT1, 3, 5a, and 5b to DNA was detected using an enzyme-linked immunosorbent assay (ELISA) specific for the respective protein-DNA-complexes. Compounds **3**, **10**, and **16** inhibited formation of the DNA complex of STAT5a and STAT5b, but not of STAT1 and STAT3 (Fig. 4b). Likewise, inhibition of the STAT5:DNA complex by compound **10** was detected in the electro mobility shift assay (EMSA) (Fig. 4c). The specificity of compound **10** was further investigated in complex cell lysates using cellular thermal shift assays (CETSA) (Fig. 4d)[35]. Compound **10** bound to isolated MBP-STAT5b-SH2 shifting the melting temperature ($T_m$) by 9 °C (Fig. 4a). BaF3/FLT3-ITD cell lysates were incubated with **10** and exposed to a temperature gradient from 43–73 °C[36]. Supernatants were analyzed by gel electrophoresis and immunoblotting was performed with STAT5a/b antibodies. Inhibitor **10** shifted the melting temperatures of STAT5a/b by 5 and 8 °C, respectively, indicating the thermal stabilization of STAT5 proteins through ligand binding in cell lysates (Fig. 4d, e).

The specific interaction of inhibitor **10** with STAT5 was further challenged by interfering with the photo-crosslinking of STAT5b-SH2 and the dual-labeled STAT5-binding peptide 5-CF-K(biotin)

GpcFLSLPPW-NH$_2$ **27** (CF = carboxyfluorescein, pcF = phos-phono-carboxy-phenylalanine)[21]. Peptide **27** was demonstrated to photo-crosslink STAT5b after being exposed to UV irradiation at 365 nm and **10** suppressed the photo-crosslinking in a concentration-dependent manner (Supplementary Figure 4). When peptide **27** was incubated with BaF3/FLT3-ITD cell lysate, irradiated for 15 min at 4 °C and subjected to pull-down using avidin beads, compound **10** repressed STAT5-crosslinking by displacing peptide **27** competitively in the complex lysate (Fig. 4f).

**Validation of STAT5 inhibitors in living cells and animals.** The biological activity of inhibitors was studied in a cellular disease model using the murine pro-B-cell line BaF3 stably transfected with the internal tandem duplication (ITD) mutation of the human FLT3 receptor (FLT3-ITD)[37]. In these cells, STAT5 is constitutively phosphorylated without cytokine activation. As the FLT3-ITD mutation is found in 35% of AML patients, it can be considered as a relevant model for this disease[38].

At first, STAT5 phosphorylation at tyrosine residues (Tyr694/Tyr699) was investigated. 5-Aryl-substituted derivatives like **11–13** could not be tested in cells as they precipitated in buffer. Compounds **10**, **16**, and **18** were dissolved in dimethylsulphoxide (DMSO) (20 mM) and added to BaF3/FLT3-ITD cells. After 6 h cells were harvested to determine the level of STAT5 phosphorylation using phosphotyrosine-specific antibodies. The strongest inhibition was observed for compound **16** with > 50% reduction of STAT5 phosphorylation at 25 μM (Fig. 5a). In contrast, the phosphorylation of STAT5 was inhibited by **10** only with an $IC_{50}$ value of > 100 μM. We suspected that the low cellular activity of **10** was hampered by low cellular uptake owing to its high polarity. This suspicion was substantiated by the higher activity of ester derivative **18**, which might act as prodrug being activated by intracellular esterases[24,33,39]. A time course experiment indeed showed that compound **18** reduced STAT5 phosphorylation steadily over 10 h (Supplementary Figure 5). Both compounds **16** and **18** had no effect on the overall expression of endogenous STAT5 (Supplementary Figure 5).

Next, inhibition of gene transcription by **16** was determined. BaF3/FLT3-ITD cells transfected with a dual firefly/Renilla luciferase system were treated with **16** for 6 h and the activity of the STAT5-transcribed luciferase reporter gene was found to reduce significantly in a dose-dependent manner (Fig. 5b). Inhibition of the endogenous transcription of STAT5 target genes by **16** and **18** was studied, too. BaF3/FLT3-ITD cells were treated with inhibitors for 6 h, mRNA was harvested and analyzed by quantitative RT-PCR. Transcription and protein expression of three target genes of STAT5, Pim1 kinase, Bcl-xl, and Cis, which play essential roles in cell cycle progression and survival, was found to be strongly reduced (Fig. 5c, Supplementary Figure 5).

The effect of STAT5 inhibitors **10**, **16**, and **18** on the proliferation of cancer cells carrying the common FLT3-ITD mutation after 48 h was quantified by the Alamar Blue assay (Fig. 5d). All three compounds showed a clear dose-dependent inhibition of cell proliferation. For comparison, the compounds were tested with four non-STAT5-dependent cell lines (HT-29, COS-7, HeLa and MDA-MB-231) and here cytotoxicity was negligible at up to 500 μM (Supplementary Figure 6). The compounds also showed no effects on STAT3 phosphorylation and on endogenous STAT3 expression (Supplementary Figures 5 and 6)[25]. The percentage of necrotic vs. apoptotic cells death after treatment with **16** was studied by flow cytometry staining with a fluorescent annexin-V conjugate and with propidium iodide (PI), likewise reduced STAT5 phosphorylation was studied using a

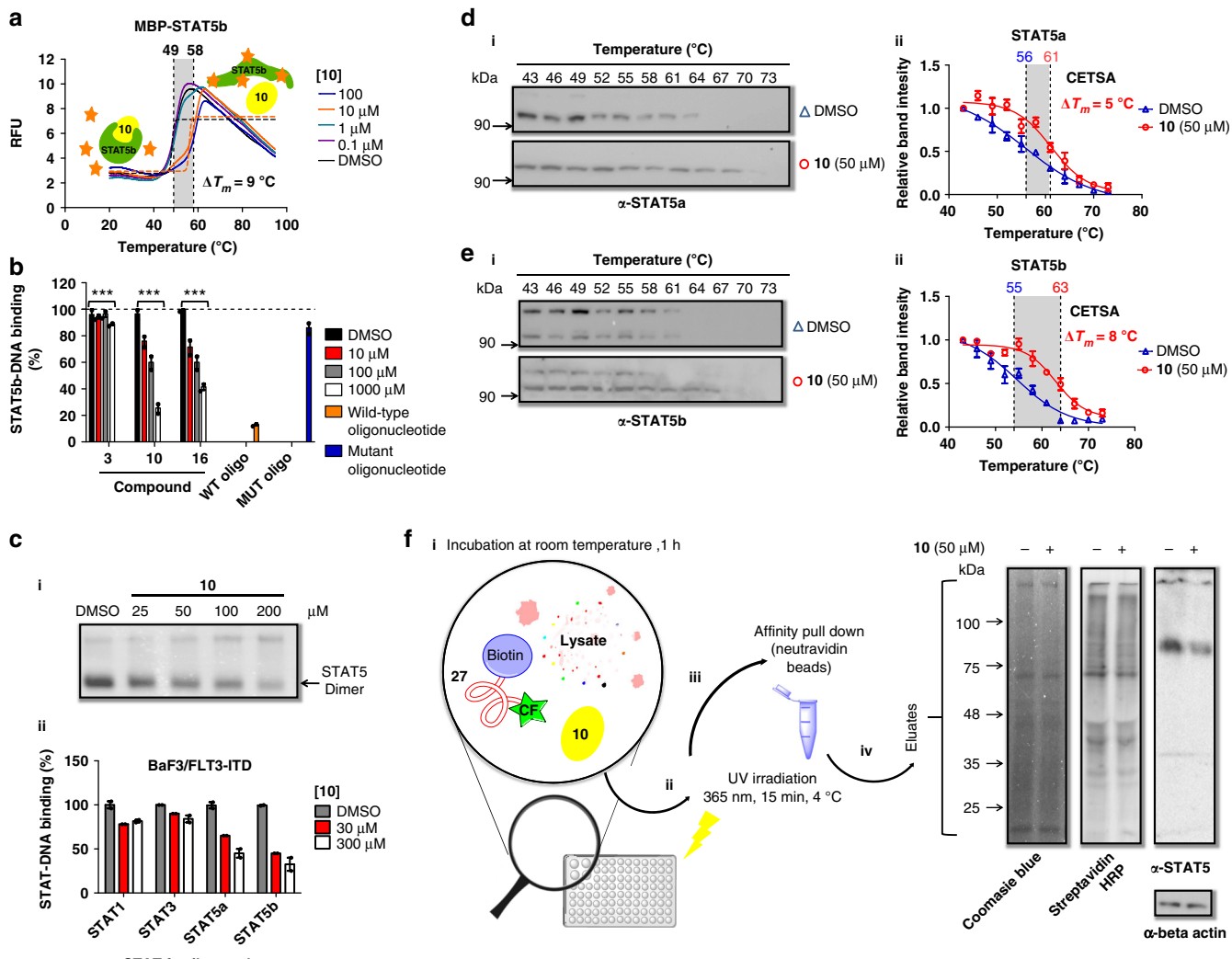

**Fig. 4** Inhibitor **10** blocks STAT5 dimerization, DNA-binding, and is active in CETSA and photocrosslinking assays in cellular lysates. **a 10** stabilizes MBP-STAT5b-SH2 shifting the melting temperature ($\Delta T_m$) by 9 °C in the TSA (data shown are one representative example of $n = 3$). **b 10** and **16** inhibit formation of trimeric (STAT5b)$_2$-DNA complexes in an ELISA ($n = 3$). **c** (**i**) **10** inhibits binding of STAT5 dimers, isolated from nuclear extracts of BaF3/ FLT3-ITD cells, to its target DNA in the electro mobility shift assay (EMSA) and (**ii**) shows selectivity for disrupting STAT5a,b-DNA complexes in the TransAM® STAT family ELISA ($n = 3$). **d–e 10** binds to STAT5a and STAT5b in complex cellular lysates as demonstrated by cellular TSA (CETSA), resulting in shifted melting curves at 50 μM compared with vehicle (DMSO). Relative STAT5a and STAT5b band intensities were plotted against corresponding incubation temperatures and fitted to Boltzmann sigmoidal curve. The blue triangle represents DMSO and red circle represents compound **10** ($n = 3$). **f** Binding of **10** to STAT5 in BaF3/FLT3-ITD cell lysate was determined by photo-crosslinking ($n = 3$). (**i**) The dual-labeled (carboxyfluorescein and biotin) peptide probe **27** binds to STAT5 with submicromolar affinity and photocrosslinked (**ii**) with target proteins by activating the 4-phosphoncarbonyl residue that acts as a photoactive phosphotyrosine-mimetic. (**iii**) Cross-linked proteins can be isolated by biotin pull-down using Neutravidine beads. (**iv**) Displacement of **27** (100 μM) by compound **10** (50 μM) in BaF3/FLT3-ITD cell lysates (1 mg ml$^{-1}$) resulted in significantly reduced photo-crosslinking of **27** and STAT5 as demonstrated in the western blotting using STAT5 antibodies (right lane), whereas other biotinylated proteins were not reduced (middle lane). For an uncropped image of **c**, **d**, **e**, **f** see Supplementary Figure 10. Error bars denote mean ± S.D. and $p$ values are considered as follows: *$p$ value < 0.05; **$p$ value < 0.01; and ***$p$ value < 0.001. Statistical analyses were performed using one-way ANOVA or the two-tailed Student's $t$ tests where appropriate

fluorophor-conjugated anti-pSTAT5 antibody (PE-Cy7 Mouse anti-STAT5, pY694, Fig. 5e).

Target specificity of compound **16** to STAT5 in living cells was evaluated using isothermal dose–response fingerprints (ITDRF)[40], a variation of CETSA experiments. BaF3/FLT3-ITD cells were treated with **16** at concentrations between 0 and 100 μM for 6 h. All samples were heated for 3 min to 60 °C, the denaturation temperature based on the $T_m$ curves in the CETSA experiment (Fig. 4d, e), lyzed and immunoblotted with STAT5a/b antibodies. The amount of STAT5 protein found in the blot was plotted

against logarithmic concentration of the inhibitor, indicating the in-cell occupancy ($OC_{50}$) of STAT5a/b of 63 and 28 μM, respectively, correlating well with the inhibition of target phosphorylation and cell proliferation (Fig. 5f, g).

Moreover, the synergistic effects of STAT5 inhibitor **16** with the staurosporine-derived FLT3-inhibitor PKC412 were investigated[41,42]. The receptor tyrosine kinase FLT3 and especially the constitutively activated mutant FLT3-ITD have been described to be responsible for the phosphorylation and, thereby, overactivation of STAT5. Treatment of AML patients

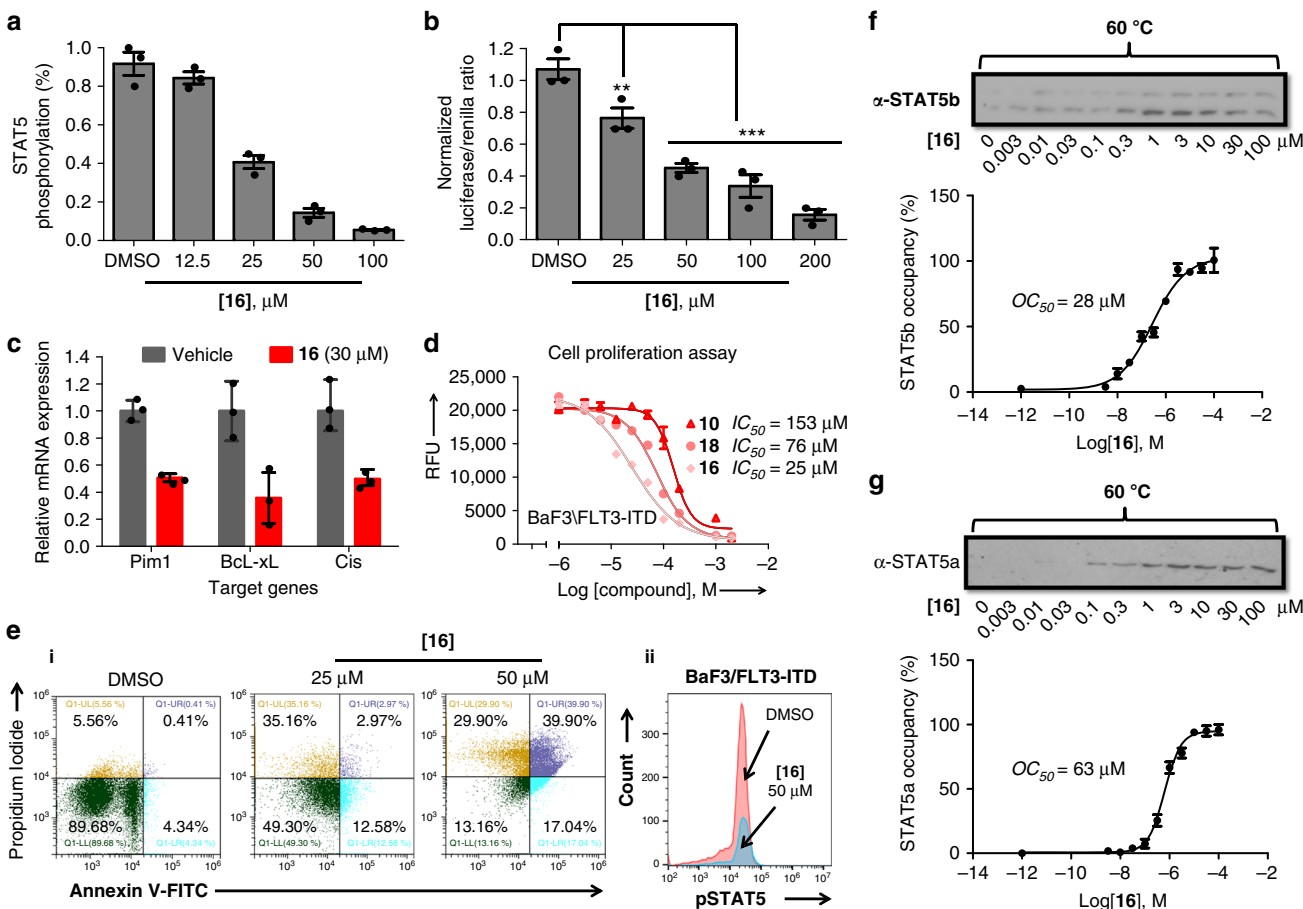

**Fig. 5** Activity, target occupancy, and functional effects of STAT5 inhibitor **16** tested in STAT5-dependent cells. **a 16** blocks tyrosine phosphorylation of STAT5 in a dose-dependent manner as shown by western blot analysis in BaF3/FLT3-ITD cells after 6 h treatment. Relative STAT5 phosphorylation levels were plotted as after quantification using Image J software. Immunoblotting for beta-actin was used as a control for uniform protein loading ($n = 3$). **b** Compound **16** inhibits transcriptional activity of STAT5 in BaF3/FLT3-ITD cells as measured by normalized Fluc/Rluc ratio in dual luciferase reporter assay ($n = 3$). **c** Expression of downstream targets of STAT5 Pim1, BcL-xL and Cis was reduced after 18 h of treatment with compound **16**. Gene expression was quantified by quantitative PCR ($n = 3$). **d** Compound **16** inhibits the proliferation of BaF3/FLT3-ITD cells after 48 h as determined by the Alamar Blue assay ($n = 3$). **e**, **i** BaF3/FLT3-ITD cells were treated with compound **16** after which annexin-V/propidium iodide staining and flow cytometry were performed (gating strategy as in Supplementary Figure 11a, data shown are one representative of $n = 3$); (**ii**) Intracellular levels of phosphorylated STAT5 were evaluated by flow cytometry after 6 h exposure of cells to compound **16** for 50 μM (gating strategy as in Supplementary Figure 11b, data shown are one representative of $n = 3$). **f–g**, In-cell occupancy of STAT5a and STAT5b by compound **16** in BaF3/FLT3-ITD, determined using ITDRF. ITDRF of compound **16** on STAT5a and STAT5b denaturation at 60 °C for 3 min based on raw data from western blotting chemiluminescence readings ($n = 3$). CETSA, cellular thermal shift assay; ITDRF, isothermal dose–response fingerprint; $OC_{50}$, the concentration at which 50% of the STAT5 in the cell was occupied by inhibitor. For an uncropped image of panels **f** and **g** see Supplementary Figure 10. Error bars denote mean ± S.D. and $p$ values are considered as follows: **$p$ value < 0.01; and ***$p$ value < 0.001. Statistical analyses were performed using one-way ANOVA or the two-tailed Student's $t$ tests where appropriate

bearing the FLT3 mutation with PKC412 alone exhibited a therapeutic effect, however, often led to major drawbacks such as incomplete target inhibition and short-lived responses. Therefore, combination treatment of leukemic cells with two inhibitors, one targeting FLT3 and the other STAT5, can be a promising strategy to enhance inhibition of STAT5 signaling, overcome or prevent resistance toward kinase inhibitors and improve treatment efficacy[38,41]. MV-411 leukemic cells were treated with concentrations of inhibitors resulting in 20% of cell apoptosis individually as well as in combination. The combination of **16** and PKC412 resulted in a threefold increase in annexin-V-staining corresponding to 60% apoptotic cells, in decreased reporter gene expression, and in reduced STAT5 phosphorylation (Fig. 6a–d). Both compounds impaired

synergistically the cell proliferation after 48 h of treatment as demonstrated by a Chou–Talalay combination index (CI) plot (Fig. 6c)[43]. For example, the $IC_{50}$ of compound **16** was reduced threefold in the presence of 1.25 nM ($IC_{10}$) of PKC412.

Finally, the inhibitory effect of **16** was examined in a murine xenograft model of leukemia. Treatment of nude mice with **16** was tolerated well and had no significant effect on mice body weight (Fig. 6e). Nude mice were inoculated subcutaneously with BaF3/FLT3-ITD cells. The control group displayed rapid tumor growth, while tumor growth in the group treated s.c. with compound **16** (200 mg kg$^{-1}$) was delayed and became first apparent on day 11 (Fig. 6f). Tumor growth in the treated group was reduced to 6% T/C on day 11 and 28% on day 14 proving that compound **16** exhibited anti-tumor efficacy in vivo.

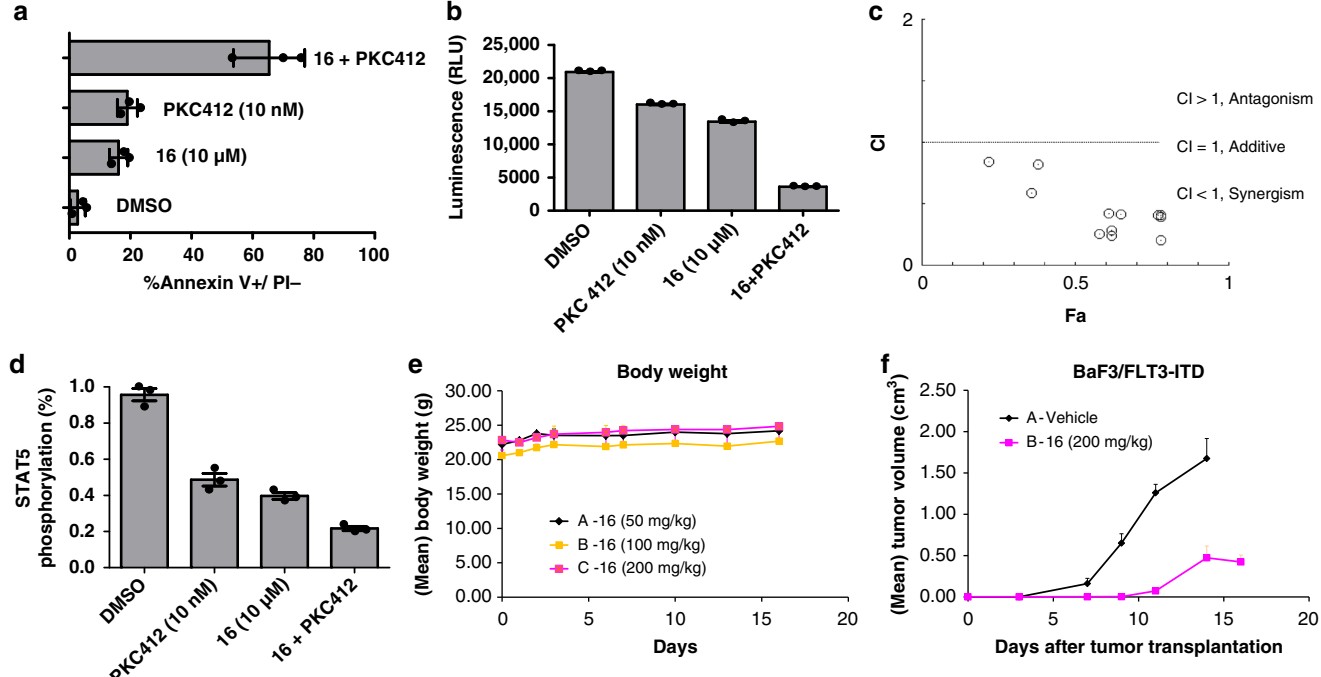

**Fig. 6** Synergy of inhibitor **16** and kinase inhibitor midostaurin (PKC412). Activity tested in a murine cancer model. **a** MV-411 cells were treated with PKC412 (10 nM) or compound **16** (10 μM) alone or in combination and incubated for 24 h followed by annexin-V/ propidium iodide staining and flow cytometry. Apoptosis was quantitated for three independent experiments ($n = 3$) and error bars denote mean ± S.D. **b** Cell viability assays were carried out by treating MV-411 cells with compound **16** (10 μM) and PKC412 (10 nM) alone or in combination. The number of viable cells was distinguished using an ATP-dependent bioluminescence assay (CellTiter-Glo, Promega) ($n = 3$, error bars denote mean ± S.D.). **c** Combination index (CI) plot showing the synergistic effect of compound **16** and PKC412 in MV-411 cells. CI values were generated using CalcuSyn software (Conservion, Ferguson, MO) and plotted as a function of fractional growth inhibition ($n = 3$) (Fa) where $Fa = (A_{570}$ control$-A_{570}$ treated$)/A_{570}$control. CI values of < 1, = 1, and >1 indicate synergism, additivity, and antagonism, respectively. Error bars denote mean ± S.D. **d** Relative STAT5 phosphorylation levels were plotted based on the raw data from western blotting chemiluminescence readings to study the synergistic effect of both compounds on STAT5 phosphorylation reduction. MV-411 cells were treated with compound **16** or PKC412 alone or in combination and incubated for 6 h ($n = 3$, error bars denote mean ± S.D.). **e** The corresponding body weight changes in non-xenografted mice during compound **16** treatments ($n = 6$, error bars denote mean ± S.D.). **f** Compound **16** significantly inhibits tumor growth in BaF3/FLT3-ITD xenograft tumor model. Time course of tumor growth suppressed by compound **16** (200 mg kg$^{-1}$) in mice bearing BaF3/FLT3-ITD tumor ($n = 6$, error bars denote mean ± S.D.)

## Discussion

In this contribution, protein-induced Mannich ligations have been discovered and characterized as protein-catalyzed, non-enzymatic reactions following a fundamental mechanism for the formation and identification of protein ligands. The reactions delivered potent, specific, and cellularly active inhibitors of the transcription factor STAT5. The starting point, phosphate-mimetic fragment **3** was activated by the protein STAT5b-SH2 enabling three-component reactions with FA and various 1*H*-tetrazoles in aqueous physiological buffer yielding low-micromolar and submicromolar inhibitors of the protein–protein interaction site. The progress of the observed Mannich ligations depended on the pH of the solution and on the presence of the protein. At pH 7.4, the reaction occurred only in the presence of the protein, whereas at pH 5.0 it was switched to an entirely protein-independent reaction. Protein-induced reactions were analyzed using FP and TS assays; products formed were quantified by HPLC-MS and protein-induced product was saturated at 432 nM (average of three independent experiments) with 250 nM protein. Product saturation is typical for protein-dependent reactions and resulted from auto-inhibition by the products formed. The reaction was also inhibited competitively by alternative ligands of the SH2 domain confirming that the phosphotyrosine recognition site constituted the catalytic center.

Remarkably, all initial 1*H*-tetrazole fragments like compound **25** did not bind to the STAT5b-SH2 protein at all ($K_I > 10$ mM),

whereas, for example, the ligation product of **3**, FA, and **25**, namely 4-(1*H*-tetrazol-1-yl)-methylamino-furazan-3-carboxylate **10**, displayed super-additive binding with an affinity of 1.4 μM and the enhanced ligand efficiency of 2.23 kJ mol$^{-1}$ per non-hydrogen atom. This findings excluded the mechanism of a protein-templated reaction, which requires the binding of both fragments to the protein template in order to initiate the reaction. Instead, an alternative mechanism of protein-dependent reactions was observed, in which only one of the starting molecules, here fragment **3**, binds to the protein. Molecular modeling suggested that H-bonding by the side chain amide of Asn642 induced the reaction of fragment **3** with FA, that of the intermediary for-miminium ion with tetrazoles, and the high selectivity of the reactions[21]. This hypothesis was challenged by investigating the protein-dependent reaction of **3**, FA, and 1*H*-tetrazole with the phosphotyrosine binding site of STAT3-SH2, which is structurally closely related containing Arg618 and Ser622 but lacks Asn642. Indeed STAT3-SH2 did not catalyze the Mannich ligation reaction and no product was identified in HPLC-MS. In agreement with this result, STAT3, STAT1, and the protein tyrosine phosphatase SHP2, all lacking the Asn-residue in the phosphotyrosine recognition sites, did not bind the STAT5 inhibitors **10** and **16**, although cross-reactivity has been reported for other STAT-inhibitors[44,45]. The importance of Asn642 was further confirmed by the mutant STAT5b-N642A, which was generated by site-directed mutagenesis. The mutant protein

displayed strongly reduced binding affinity to peptide **1** and to inhibitor **10** and did not catalyze a Mannich ligation reaction as shown for the wildtype protein. In conclusion, the observed protein-dependent reaction constitutes indeed a protein-induced reaction in which the protein catalytically activates the bound fragment **3** by specific binding interactions without templating the reacting fragments priorly.

Specificity of the formed inhibitors for STAT5 was also confirmed in complex cellular systems. Binding of inhibitors **10** and **16** to STAT5a,b in cell lysates was determined by CETSA. The high selectivity of the protein-ligand interaction was demonstrated by the fact that inhibitor **10** in cell lysates suppressed photo-crosslinking of the biotinylated STAT5 ligand **27** with STAT5 but not with other proteins. In addition, the selectivity of compound **16** in living cells was proven by ITDRF. It was found that 50% of the target proteins STAT5a and b were occupied in living cells with **16** at concentrations ($OC_{50}$ values) being in the same range as the $IC_{50}$ values observed for the inhibition of STAT5 phosphorylation and cell proliferation, indicating that the interaction of **16** with the target STAT5 alone was sufficient to generate and explain the observed biological effects. Likewise, compound **16** was able to block STAT5 transcription and to inhibit cellular proliferation of cancer cells in a mouse model. Another proof for the cellular selectivity of **16** for STAT5 was that the proliferation of all tested cell lines which proliferate independently from STAT5 activation was not inhibited by **16** and these cells showed no sign of toxicity.

Our work demonstrates the growing power of protein-dependent fragment ligations in fragment-based drug discovery. Protein-induced multicomponent reactions such as Mannich ligations enlarge the chemical diversity of protein ligands accessible by the method considerably. Protein-induced fragment ligations seem to be especially advantageous for the formation of potent and specific ligands as molecular interactions responsible for binding catalyze the ligation reaction. Considering the omnipresence of FA in living cells[46,47] and the versatility of reactions this and other aldehydes can undergo, the mechanism should find broad application on numerous protein targets and many bioactive fragments should be expandable to chemically diverse and potent protein ligands, possibly even in living cells. Thus, protein-induced reactions seem to constitute an additional, non-enzymatic mechanism exerted by proteins enabling the molecular evolution of ligands and modulating protein activities and functions.

## Methods

**General methods.** Protocols and reaction schemes for the chemical synthesis of compounds **1–27** as well as other methodological information are reported in the Supplementary Methods. Identity and purity ( > 95%) of all compounds were determined by chromatography (silica or RP-18 HPLC), by fully assigned 1H- and 13C-NMR spectra, by high-resolution mass spectra and by elemental analysis.

**Antibodies and reagents.** The following antibodies were used for immunoblotting: anti-STAT5a (C-6, cat. sc271542, 1 µg ml$^{-1}$) and STAT5b (G-2, cat. sc-1656, 1 µg ml$^{-1}$), anti-STAT3 (C-20, cat. sc-482, 1 µg ml$^{-1}$), anti-phospho-STAT3 (Tyr 705) (cat. sc-7993, 1 µg ml$^{-1}$), anti-STAT5 (C-17, cat. sc-835, 1 µg ml$^{-1}$), and anti-Bcl-xL (H-5, cat. sc-8392, 1 µg ml$^{-1}$) were obtained from Santa Cruz Biotechnology (Santa Cruz, CA). Antibodies recognizing Cyclin D1 (92G2, cat. 2978 s, 2 µg ml$^{-1}$), FLT3 (8F2, cat. 3462 s, 1 µg ml$^{-1}$), phospho-STAT5 (Tyr694) (cat. 9351, 1 µg ml$^{-1}$), phospho-FLT3 Tyr591 (cat. 3461 s, 1 µg ml$^{-1}$) were obtained from Cell Signaling Technology (Danvers, MA). Anti-Pim1 (M08, clone 1C10, cat. H00005292-M08, 2 µg ml$^{-1}$) was purchased from Abnova (Taiwan), anti-Beta-actin (cat. ab8227, 0.5 µg ml$^{-1}$) was from Abcam (Cambridge, MA), anti Biotin−Peroxidase antibody (cat. A4541, 1 µg ml$^{-1}$) was obtained from Sigma-Aldrich (Taufkirchen, Germany) and anti-CIS (cat. PA5−27128, 1 µg ml$^{-1}$) was from Thermo Fisher Scientific (Braunschweig, Germany). For phospho flow cytometry experiments, PE-Cy™7 mouse anti-Stat5 (pY694) (clone 47, cat. 560117, 20 µl each test) and anti-mouse IgG1 monoclonal antibody (clone A85-1, cat. 550083, 0.2 µg ml$^{-1}$) were from BD Biosciences. Antibodies were validated by the

manufacturers. Recombinant mouse IL3 protein (cat. PMC0034, 1 ng ml$^{-1}$) was purchased from Thermo Fisher Scientific and midostaurin hydrate (PKC412, cat. M1323) was from Sigma-Aldrich. Both TransAM STAT family kit (cat. 42296) and nuclear extraction kit (cat. 40010) were from Active Motif (Carlsbad, CA, USA).

**Cell lines and cell culture.** MV-411 (CRL-9591) and K562 (CCL-243) cells were from American Type Culture Collection (ATCC), (LGC Standards GmbH, Wesel, Germany). BaF3 and BaF3/FLT3-ITD cells were kind gifts from professor Carol Stocking (University Hospital Hamburg-Eppendorf). All suspension cells were cultured in Rosewell Park Memorial Institute media supplemented with 10% fetal calf serum (FCS) without or without recombinant mouse IL3 (1 ng ml$^{-1}$). COS-7 (CRL-1651), HeLa (CCL-2), MDA-MB-231 (HTB-26), and HT-29 (HTB-38) were from and certified by ATCC and cultured in DMEM media supplemented with 10% FCS. Cell culture media and FCS were purchased from Biozol (Biozol, Eching, Germany). Cells were incubated at 37 °C, 5% CO$_2$ atmosphere. All cell lines were tested and were mycoplasma-free using MycoAlert Mycoplasma Detection Kit (Lonza).

**Detection of protein-induced ligand formation via FP assay.** All FP assays were conducted in a total volume of 20 µl in 50 mM MOPS buffer (pH 7.4) at room temperature. For investigating the tolerance of the assay for FA, serial dilutions of FA (concentration range: 0–1 mM) were prepared and incubated with 250 nM of MBP-STAT5b in buffer. For the protein-induced reaction, 250 nM of MBP-STAT5b protein were added to a mixture of 250 µM of **3** and one heteraryl nucleophile per microtiter plate well with increasing concentration of FA up to the concentration of 250 µM. Reaction mixtures were incubated for 12 h with mild shaking. Plates were centrifuged and 10 nM of FP probe **1** were added and incubated for 1 h with mild shaking before measurement with Safire[2] well plate reader (Tecan, Crailsheim, Germany).

**Detection of protein-induced ligand formation via TSA.** Here we introduced the possibility of using TSA[48] assay to determine protein-induced ligand formation. The compatibility of FA with the TSA was investigated using MBP-STAT5b-SH2 protein (500 nM) mixed with increasing concentrations of FA up to the concentration of 250 µM and incubated at room temperature with mild shaking in a sealed 384 PCR plate. To determine the formation of ligand via a protein-induced reaction, MBP-STAT5b-SH2 protein 500 nM was added to 250 µM (IC$_{20}$) 4-amino-furazane-3-carboxylic acid **3**, equimolar amount (250 µM) of one heteraryl nucleophile per PCR plate well with increasing concentration of FA up to the concentration of 250 µM. Reaction mixtures were incubated for 12 h with mild shaking at room temperature, respectively. After 12 h, the plates were centrifuged and 1 µl of 400X Sypro Orange solution (Thermo Scientific) was added, resulting in a total assay volume of 20 µl, with a final protein concentration of 475 nM. The PCR plates were again sealed with optical seal, shaken for 15 min, and centrifuged. Thermal scanning (20–95 °C at 1 °C min$^{-1}$) was performed using a real-time PCR setup (LightCycler (Roche Diagnostics, Mannheim, Germany) and fluorescence intensity was measured after every 0.3 s. Curve fitting, melting temperature calculation, and report generation on the raw data were performed using GraphPad Prism 5 software.

**Detection of protein-induced ligand formation by LC/MS.** Extracted ion chromatography was performed with reaction mixtures containing 250 nM of MBP-STAT5b protein, 250 µM (IC$_{20}$) 4-amino-furazane-3-carboxylic acid **3**, equimolar amount (250 µM) of one heteraryl nucleophile and FA with a total volume of 100 µL. The reaction mixtures were vortexed to mix thoroughly and incubated overnight at room temperature and was analyzed using a HPLC/QTOF-MS instrument by Agilent, consisting of an Infinity 1290 UHPLC coupled to a 6550 iFunnel QTOF. After 12 h each sample was analyzed in triplicate by injecting (10 µl) into the LC/MS instrument and the ligation products were identified by their molecular weights and by comparison of the retention times of synthetic reference. Calibration curves for hit compounds **10** and **16** are given in Supplementary Figure 19.

**Determination of STAT-DNA binding using ELISA.** Binding of STAT1, STAT3, STAT5a, and b to DNA was determined from nuclear protein extracts using the TransAM STAT family kit from Active Motif (Carlsbad, CA). Nuclear protein extract was prepared by resuspending nuclear pellet by pipetting up and down in 50 µl of complete lysis buffer (Active Motif, Carlsbad,CA) in the presence of 2.5 µl of detergent to ease solubilization of membrane associated nuclear proteins. Suspension was incubated on ice for 30 min on a rocking platform at 150 rpm followed by 30 s vortex at highest setting. Nuclear extract fraction was collected through centrifugation at 14,000×g and protein concentration was determined using Bradford assay. Nuclear extract (2 µl of a 1 µg µl$^{-1}$ solution) containing activated, dimeric STAT was dissolved in a mixture of a solution of the inhibitor **3**, **10**, or **16** in complete lysis buffer (20 µl, 1% (v/v) DMSO) and complete binding buffer (30 µl) in microtiter wells coated with DNA of the respective STAT consensus sequence. After incubation at room temperature for 1 h, the wells were washed three times with the wash buffer and incubated with respective STAT antibodies for 1 h. After washing as before, wells were incubated with a secondary antibody

conjugated to horseradish peroxidase for 1 h at room temperature. Wells were washed again for three times, incubated with 100 μl of developing solution, quenched with 100 μl of the stop solution, and the absorbance was measured at 450 nm using Safire² well plate reader (Tecan, Crailsheim, Germany).

**EMSA**. Gel shift assay was conducted using a double-stranded, biotin-labeled oligonucleotide probe containing the consensus binding site for STAT5 (sense strand, 5′-AGATTTCTAGGAATTCAATCC-3′), using the Gelshift Chemiluminescent EMSA kit (Active Motif). In brief, protein–DNA complexes were resolved on a nondenaturing polyacrylamide gel, transferred to a positively charged nylon membrane, and cross-linked to the membrane using the UV-light cross-linker. After blocking, the membrane was incubated with blocking buffer containing streptavidin conjugated to HRP. After washing, protein–DNA complexes were detected using a chemiluminescent substrate (Active Motif, Carlsbad, CA, USA)[40,49].

**Photo-crosslinking and enrichment of biotinylated proteins**. Peptide probe **27** (100 μM) was incubated with BaF3/FLT3-ITD whole cell lysates (1 ml of 1 mg ml⁻¹) in the binding buffer (50 mM HEPES, pH 7.5, 200 mM NaCl, 2 mM MgCl₂, 0.1% Tween-20, 20% glycerol, 2 mM phenylmethylsulfonyl fluoride, Roche Complete ethylenediaminetetraacetic acid-free protease inhibitor cocktail) for 1 h at 4 ℃. The samples were then irradiated at 365 nM using a UV transilluminator for 15 min at 4 ℃. For competitive displacement studies, compound **10** (50 μM) was added together with peptide probe **27** and incubated with whole cell lysates for 1 h at 4 ℃ prior to UV photo-crosslinking for 15 min at 4 ℃. The mixture was then incubated with Neutravidin agarose resin (Thermo Fisher Scientific) for 2 h at room temperature. After washing with phosphate-buffered saline with 0.1% sodium dodecyl sulfate (SDS), the enriched proteins were eluted by boiling the beads in SDS-polyacrylamide gel electrophoresis sample buffer.

**Statistical analysis**. Statistical calculations were performed using GraphPad Prism 5.01 software and reported as mean ± S.D. Experiments were performed in triplicates and/or repeated at least three times unless indicated otherwise. Two-tailed Student's $t$ tests and one-way analysis of variance were used to identify statistically significant data. $p$ values are considered as follows: *$p$ value < 0.05; **$p$ value < 0.01; and ***$p$ value < 0.001 and statistical significance was attributed to $p$ values < 0.05. Synergy in cell viability assays was determined by plotting isobolograms and calculating the CI using CalcuSyn software (Calcusyn software, Biosoft, San Diego, CA, USA) (Conservion, Ferguson, MO) using the Chou–Talalay method[43] to ascertain if the effects of drug combinations were synergistic ($CI < 1$), additive ($CI = 1$), or antagonistic ($CI > 1$).

**Animal experiments**. NSG mice (NOD/Shi-*scid*/IL-2Rγ^null) obtained from The Jackson Laboratory aged 6 weeks with an average body weight of 22 g. All animal experiments have been approved by the Tierversuchskommission (commission for animal experiments) of the Landesamt für Gesundheit und Soziales (LaGeSo, the state office for health and social affairs), which is the responsible ethics committee for all animal experiments conducted in the state of Berlin. All animal experiments were carried out in accordance with the United Kingdom coordinating committee on cancer research regulations for the welfare of animals and with German Animal Protection Law and were reviewed by the animal protection officer (Tierschutzbeauftragter) at Experimental Pharmacology & Oncology (EPO) Berlin-Buch GmbH. BaF3/FLT3-ITD cells were injected into mice subcutaneously (10⁶ cells per mouse). Inoculated and control mice (six in each group) were treated for 16 d once daily s.c. with either the vehicle (10% (v/v) DMSO/0.25% Tween 80) or compound **16** (200 mg kg⁻¹) dissolved in vehicle, starting shortly after tumor cell inoculation. Former studies had shown that this concentration of test compound is well tolerated by the mice. Tumor volumes and body weights were recorded daily ($n = 6$, error bars denote mean ± S.D.).

**Reporting Summary**. Further information on experimental design is available in the Nature Research Reporting Summary linked to this article.

## Data availability statement

The authors declare that the data supporting the findings of this study are available within the article and its Supplementary Information files. A reporting summary for this Article is available as a Supplementary Information file. The source data underlying Figs. 3b, 3d, 3f, 4a, 6e, f, and Supplementary Figures 2g–i have been submitted to FigShare: https://figshare.com/s/4a8e28d4337b5509c6ed. All other relevant data supporting the findings of this study are available from the corresponding author upon request.

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

## Acknowledgements

Support is gratefully acknowledged from the Investitionsbank Berlin (grant No. 10142914), the Deutsche Krebshilfe, and the DFG (FOR 806, SFB765, and the Berlin School of Integrative Oncology for a fellowship granted to E.L.W.). We thank Professor Thorsten Berg, University of Leipzig, for the plasmid containing STAT5-MBP and Professor Carol Stocking, University Hospital Hamburg-Eppendorf for providing us with the BaF3/FLT3-ITD cell line. We are grateful to Dr. Anne Diehl, FMP, for protein expression, to Dr. Jens Peter von Kries, FMP, for screening support, to Dr. María Isabel Fernández-Bachiller, FU Berlin, for initial discussion on FP assays, and to Thomas Rudolf, FU Berlin, for supporting chemical synthesis.

## Author contributions

J.R., E.L.W., D.C., and C.A. conceived and designed the experiments, E.L.W., B.G.K., S.B., and X.W. performed the experiments. E.L.W., C.A., M.L., and J.R. analyzed the data. E.N., B.G.K., S.B., C.A., and S.W. contributed reagents. J.R. and E.L.W. wrote the manuscript. J.R. acquired funding for and administered the project.

## Additional information

**Competing interests:** The authors declare no competing interests.

**Journal Peer Review Information:** *Nature Communications* thanks the anonymous reviewers for their contribution to the peer review of this work

