## [Peer Review File · Nature Communications]

Reviewers' comments:

Reviewer #1 (Remarks to the Author):

Wong et al

The manuscript focuses on protein catalyzed Mannich ligation as an approach to discover small molecule inhibitors of Stat5. In these studies, Stat5 serves as the template that catalyzes three component reactions of phosphate mimetic, formaldehyde, and 1 H-tetrazoles to generate ligands with binding affinity for the protein. Sufficient details of the chemistry and the mechanics of the reaction and its conditions are provided. The significant reaction products that bind and serve to inhibit Stat5 activity include compounds 3, 10, and 16. The authors then proceed to characterize the activities and the properties of the three leading compounds in regard to the Stat5 inhibition and the effects on Stat5, the specificity, and the effects on leukemia cell phenotype in vitro and the development of subcutaneous leukemia models in vivo. The activities of the three compounds are all micromolar, in the range of 10-150 micromolar, while the cellular activities of compound 10 appear to be limited by its poor physicochemical properties. Specificity and in vivo activity are well demonstrated. Overall, the manuscript is well written, the studies are well conducted and described, and they contain the appropriate controls for the most part.

There are issues that need to be addressed before the manuscript is acceptable for publication.

1. Molecular explanation for why the combination studies of compound 16 with staurosporine had synergistic effect on Stat5 phosphorylation (page 10, lines 229-238).
2. Some explanation for the inhibition of intracellular Stat5 phosphorylation, supported by some studies to show the mechanism of inhibition
3. Some rescue experiment with constitutively-active Stat5 that cannot be inhibited by the compounds and to show that the exogenous expression of such a construct makes the cells refractory to the effects of the inhibitors
4. The tumor model and its relevance to human leukemia, given that the model used is more artificial. There are human lines, such K562 cells, which harbor constitutively-active Stat5 that could be used to further validate the study

Reviewer #2 (Remarks to the Author):

Suggestion: Major revisions

As an addition to the field of fragment-based drug discovery and protein-dependent fragment ligation reactions, the authors present a novel and innovative approach to inhibiting cancer cell proliferation – efforts focus on the oncogene, STAT5b.

Comments

1. This research relies on a protein forming a Mannich ligation product in vivo. This reaction which requires a primary or secondary amine, an aldehyde, formic acid (introduced by the researcher or found in cellulose) and an acidic residue (Asn642, STAT5b). There seems to be a lack of selectivity for any particular protein binding pocket. The authors show that their lead fragment only interacts with STAT5 relative to other closely related STAT isoforms. In the presence of the fragment, formic acid and any acidic residue on another protein surface, there seems no reason why these Mannich ligands may not form. It would seem appropriate to either expand their discussion of selectivity across different proteins, enzymes or perform further selectivity experiments on a larger target set.

2. The authors claim residue Asn642 is critical for this protein-dependent activation. To validate this, experiments involving STAT3 were employed (which lacks this residue). No ligation was

observed as was hypothesized. This result could however be misleading given the nuances of STAT structure and complexation events. Although highly homologous and structurally similar, STAT3 is not STAT5. STAT3 and STAT5 have been shown to differentially form higher order complexes with STAT5 known to adopt STAT5 tetrameric protein-protein interactions, as opposed to STAT3's canonical dimer with itself. There may be other factors with regards to protein surface topology, SH2 domain dynamics, etc. why the ligation reaction did not work. This reviewer would suggest that the authors perform mutagenesis experiments on STAT5b and replace the Asn642 rather than using a different homolog.

3. The ligation reaction relies on the cellular availability of formic acid (FA). Although briefly mentioned by the authors, there was no detailed discussion/experimental calculation of the actual concentrations of FA in mammalian cells. While experiments were done with increasing concentrations of FA, the authors should repeat all experiments at physiologically relevant concentrations of formic acid. The authors should also comment on any variations of FA concentrations across cell types, including oncogenic vs. non-oncogenic cells. This seems a critical point for improving therapeutic index and for the future success of this approach.

4. While the inhibition of STAT5 activity is evident, the fragments utilized are structurally simple. While typically considered a positive attribute, affording oral bioavailability, cell penetrance, BBB permeability etc, from the perspective of developing inhibitors of protein-proteins interactions, the structures do not possess functionality which would be sufficient to direct binding to STAT5 selectively. It would seem likely that they might bind to many other enzymatic active sites. This should be discounted or selectively demonstrated on a wider range of protein substrates.

5. The fragments contain metabolically labile groups (prior/after the Mannich reaction), such as a carboxylate for instance. The authors should consider exploring the clearance rates of these fragments in an in vitro hepatocyte assay.

Minor Comments

- Authors used a dose of 200 mg/kg (no toxicity was observed in mouse study), however this does seem as quite a large dose, a comment on the reason behind this? Or scalability in the future?

- The Western blots presented in Figure 4 (especially, f. IV) are hard to visualize and poor quality. The bands are almost unrecognizable and inconclusive (may require a repeat).

Reviewer #3 (Remarks to the Author):

The manuscript by Rademann and colleagues reports an implementation of the protein-templated ligation in order to discover new inhibitors for STAT5 transcription factor. Such transcription factor being involved in various cancers is considered as a difficult drug target. Remarkably, in the manuscript authors report the discovery of a series of compounds that can bind to this protein and inhibit its phosphorylation, thereby affecting its function. The study is performed rigorously and the findings are of high significance. The implementation of three-component Mannich reaction to produce libraries of inhibitors is particularly innovative. Overall, the manuscript can be recommended for publication with minor modifications.

Comments:

1) On page 5, lines 101-107 the description of in situ assays for Mannich ligation is confusing. First, three components for ligation are added at pH 5, then protein MBP-STAT5b is added in buffer at pH 7.4. What is the actual pH value of the catalyzed ligation in the assays? Authors stated that background reaction (without protein) proceeds spontaneously at pH 5 and does not occur at pH

7.4, while upon addition of STAT5b reaction can occur. The current description gives an impression that pH is between 5 and 7, which may result in significant background reaction.

2) Page5, line 108, "Figure 2b": in the actual Figure 2 there is no "b", the correct designation should be added; "SData" should be "SData1" or "SData2"?

Reviewer #1 (Remarks to the Author):

The manuscript focuses on protein catalyzed Mannich ligation as an approach to discover small molecule inhibitors of Stat5. In these studies, Stat5 serves as the template that catalyzes three component reactions of phosphate mimetic, formaldehyde, and 1 H-tetrazoles to generate ligands with binding affinity for the protein. Sufficient details of the chemistry and the mechanics of the reaction and its conditions are provided. The significant reaction products that bind and serve to inhibit Stat5 activity include compounds 3, 10, and 16. The authors then proceed to characterize the activities and the properties of the three leading compounds in regard to the Stat5 inhibition and the effects on Stat5, the specificity, and the effects on leukemia cell phenotype in vitro and the development of subcutaneous leukemia models in vivo. The activities of the three compounds are all micromolar, in the range of 10-150 micromolar, while the cellular activities of compound 10 appear to be limited by its poor physicochemical properties. Specificity and in vivo activity are well demonstrated. Overall, the manuscript is well written, the studies are well conducted and described, and they contain the appropriate controls for the most part.

We are grateful for this very positive characterization of our work.

There are issues that need to be addressed before the manuscript is acceptable for publication.

1. Molecular explanation for why the combination studies of compound 16 with staurosporine had synergistic effect on Stat5 phosphorylation (page 10, lines 229-238).

PKC412 is a staurosporine-derived, potent inhibitor of the kinase domain of the receptor tyrosine kinase FLT3, which is responsible for the phosphorylation and over-activation of STAT5 leading to cellular hyperproliferation in those cases of AML carrying the FLT3-ITD mutation. Therefore, the combinatorial targeting of STAT5 and FLT3 with both a STAT5 and an FLT3 inhibitor should be a valuable strategy for AML treatment in these cells. To test this hypothesis, we investigated the functional synergism of the two inhibitors PKC412 and compound 16 acting on the two targets, FLT3 and STAT5, within the same signal transduction pathway. Our experiments revealed the synergistic inhibition of STAT5 phosphorylation and the induction of apoptosis at significantly lower doses of the two inhibitors (IC₂₀) when compared to the use of the single substances (Figure 6a-d). We have added the molecular explanation of the observed synergism to the text (page 10-11, lines 240-247).

2. Some explanation for the inhibition of intracellular Stat5 phosphorylation, supported by some studies to show the mechanism of inhibition.

STAT5 inhibitors bound to the SH2 domain of STAT5 block the recruitment of the non-phosphorylated protein to the kinases that are responsible for the phosphorylation and thereby activation of STAT5. In native "healthy" cells ligand-activated receptor tyrosine kinases (RTK) are the main drivers of STAT5 phosphorylation, whereas in cells carrying constitutively activated kinases like FLT3-ITD or BCR/ABL these mutated kinases take over the STAT5 activation. In order to develop a phosphopeptide binding to the SH2 domain of STAT5, we have earlier synthesized a small collection of phosphopeptides derived from receptor tyrosine kinases reported to activate STAT5 (ref 21). From this collection the carboxyfluorescein-labeled octapeptide 1, derived from the GM-CSF receptor (granulocyte macrophage colony stimulation factor receptor), was found to bind the STAT5-SH2 domain with highest affinity ($K_D=55$ nM) and thus was used to screen for ligands of STAT5-SH2 in a competitive fluorescence polarization assay (see SData 1). Thus, our inhibitors are ligands of the STAT5-SH2 domain and thereby block the intracellular recruitment and phosphorylation of STAT5. Further, we have shown, that the ligands block gene transcription by STAT5 and are able to inhibit the formation of the ternary DNA-STAT5-dimer complex (EMSA experiment), but not the DNA complexes of STAT1 and STAT3. We added this explanation to the text (page 3 and 4, lines 55-61), which is also elaborated in refs 25 and 34.

3. Some rescue experiment with constitutively-active Stat5 that cannot be inhibited by the compounds and to show that the exogenous expression of such a construct makes the cells refractory to the effects of the inhibitors.

Indeed, a previous study has shown that overexpression of constitutively active STAT5 can rescue leukemic cells from cell death induced by a specific STAT5 inhibitor (ref 20). We have instead used the genetic knockdown with siRNA to deplete endogenously expressed STAT5 and to verify thereby the selectivity of our inhibitors toward STAT5. These novel data have added to the SI part of the manuscript (SData 3). Genetic knockdown of STAT5 by STAT5-siRNA in the STAT5-driven leukemic cell line K562 depleted STAT5 expression in comparison to control siRNA, which had no effect (SData 3e,f). As expected, compound 16 did not affect STAT5 expression, neither in the case of STAT5-siRNA nor in that of control siRNA treatment (SData 3e,f). Cell viability was reduced by >50% when K562 cells were treated with inhibitor 16 and control siRNA, while there was no effect on viability with only control siRNA (SData 3g). On the contrary, STAT5-siRNA reduced viability by ca. 80% without any additional effect of compound 16. These results strongly suggest that the effect of compound 16 on cell viability is exerted by inhibition of STAT5 and not by additional off-target effects.

4. The tumor model and its relevance to human leukemia, given that the model used is more artificial. There are human lines, such K562 cells, which harbor constitutively-active Stat5 that could be used to further validate the study

The tumor model using murine cells is relevant to human leukemia as ca. 35% of AML patients carry the employed FLT3-ITD mutation. For further validation we have tested two additional human cell lines with constitutively active STAT5. First, MV 4-11 cells, a human acute myeloid leukemia (AML) cell line expressing the FLT3/ITD mutation and resulting in STAT5 hyper-phosphorylation was used. Compound **16** reduced cell viability and inhibited cell proliferation in MV 4-11 cell lines (IC_{50} = 28 μ M, SFigure 6a-b). In addition, STAT5 phosphorylation in MV 4-11 cells was significantly blocked with compound **16** while total STAT5 levels remained unaffected (SFigure 6c). Secondly, we have now added the results with K562 cells, a human chronic myeloid leukemia (CML) cell line bearing BCR-ABL was employed for validation (see SData 3a-d). Compound **16** potently inhibited STAT5 phosphorylation leading to reduced cell viability in K562 cell lines. Treatment of compound **16** inhibited the proliferation of K562 cells (IC_{50} = 36 μ M) showing comparable efficacy to other STAT5 driven cell lines used in our study. These findings prove that compound **16** inhibit STAT5 not only in the mouse model (BaF3/FLT3-ITD) but also human leukemic cell lines (MV 4-11 and K562) that are known to be driven by constitutively active STAT5.

Reviewer #2 (Remarks to the Author):

Suggestion: Major revisions

As an addition to the field of fragment-based drug discovery and protein-dependent fragment ligation reactions, the authors present a novel and innovative approach to inhibiting cancer cell proliferation – efforts focus on the oncogene, STAT5b.

Comments

1. This research relies on a protein forming a Mannich ligation product in vivo. This reaction which requires a primary or secondary amine, an aldehyde, formic acid (introduced by the researcher or found in cellulose) and an acidic residue (Asn642, STAT5b). There seems to be a lack of selectivity for any particular protein binding pocket. The authors show that their lead fragment only interacts with STAT5 relative to other closely related STAT isoforms. In the presence of the fragment, formic acid and any acidic residue on another protein surface, there seems no reason why these Mannich ligands may not form. It would seem appropriate to either expand their discussion of selectivity across different proteins, enzymes or perform further selectivity experiments on a larger target set.

The reported Mannich ligations are catalyzed by STAT5 protein in-vitro, not in-vivo, using an amine, formaldehyde (not formic acid) and a heterocycle as nucleophile. We have checked the manuscript for clarity in this respect. Indeed, the reaction was very specifically catalyzed by the active site of STAT5 (see Figure 3). The most important requirement for any protein binding pocket to catalyze the reaction is to bind both the starting fragment and the formed Mannich ligation product. We have tested the Mannich ligation reaction with closely related proteins (STAT3, mutated STAT5 N642A), further proteins with phosphotyrosine recognition sites (SHP2, PTP1B) and MBP as the used isolation tag and have not found the catalytic formation of the ligand in any of these cases. Furthermore, the binding of the inhibitors to proteins was investigated in whole cell lysates using cellular thermal shift assays (CETSA) and photocrosslinking assays. CETSA indicated that the binding of the inhibitors to STAT5 still occurs in the lysates, suggesting that there was no significant competition for the ligand by other proteins. The photo-crosslinking of peptide **26** with STAT5 but not with other proteins was inhibited by compound **10** as shown in Figure 4f. Finally, the selectivity of the STAT5 inhibitors was investigated in living cells using the isothermal dose response fingerprints (ITDRF) (Figure 5f,g). The cellular occupancy of STAT5a protein (OC_{50} = 25 μ M) in cells was close to the observed IC_{50} value of cellular proliferation (28 μ M), indicating that indeed binding of the inhibitors to STAT5 – and not to other targets - was responsible for the observed phenotype. In addition, the results of the siRNA experiments suggested that the effect of compound **16** on cell viability was exerted by inhibition of STAT5 and not by additional off-target effects. Furthermore, compound **16** affected cell viability only in cell lines known to be driven by STAT5, i.e. K562 (human CML cell line), MV-4-11 (human AML cell line), and BaF3/FLT3-ITD (SFigure 6a-f) while it did not exhibit cytotoxicity toward normal epithelial cells and other types of cancers that are not driven by STAT5 (SFigure 6g-k).

2. The authors claim residue Asn642 is critical for this protein-dependent activation. To validate this, experiments involving STAT3 were employed (which lacks this residue). No ligation was observed as was hypothesized. This result could however be misleading given the nuances of STAT structure and complexation events. Although highly homologous and structurally similar, STAT3 is not STAT5. STAT3 and STAT5 have been shown to differentially form higher order complexes with STAT5 known to adopt STAT5 tetrameric protein-protein interactions, as opposed to STAT3's canonical dimer with itself. There may be other factors with regards to protein surface topology, SH2 domain dynamics, etc. why the ligation reaction did not work. This reviewer would suggest that the authors perform mutagenesis experiments on STAT5b and replace the Asn642 rather than using a different homolog.

We have prepared the mutant STAT5 N642A by site-directed mutagenesis and confirmed the sequence of the mutant gene construct. The mutant protein was found to bind to the phosphotyrosine residue with reduced affinity (Kd ca. 5 μ M instead of 55 nM of the wildtype protein) (SFigure 2g). Binding of inhibitor **10** was reduced to 30 μ M (from 1.4 μ M for the non-mutated protein) (SFigure 2h). The Mannich ligation reaction with fragment **3**, formaldehyde and 1H-tetrazol conducted at pH7.4 yielded no product (Figure 3e, lane11). These results strongly support our hypothesis on the functional relevance of the residue Asn642 for the Mannich ligation reaction and for the binding of the formed inhibitors.

3. The ligation reaction relies on the cellular availability of formic acid (FA). Although briefly mentioned by the authors, there was no detailed discussion/experimental calculation of the actual concentrations of FA in mammalian cells. While experiments were done with increasing concentrations of FA, the authors should repeat all experiments at physiologically relevant concentrations of formic acid. The authors should also comment on any variations of FA concentrations across cell types, including oncogenic vs. non-oncogenic cells. This seems a critical point for improving therapeutic index and for the future success of this approach.

As mentioned in the answer to comment 1, the manuscript reports ligation reactions of formaldehyde (FA) proceeding in-vitro. We have conducted the experiments with increasing concentrations of FA finding that a concentration of 250 μ M yielded optimal results of the ligation reaction while not affecting the protein and the protein assays (FP and TSA). This concentration of 250 μ M FA is in fact a physiologically relevant concentration. According to reference 46 and 47, the concentration of FA in mammalian cells is around 400 μ M. For these reasons we mention in the conclusion that ligation reactions as those reported here might also occur in living cells, however, it was never our intention or part of this project to investigate these reactions in living cells. We would rather consider it as an interesting implication of our work for future research.

4. While the inhibition of STAT5 activity is evident, the fragments utilized are structurally simple. While typically considered a positive attribute, affording oral bioavailability, cell penetrance, BBB permeability etc, from the perspective of developing inhibitors of protein-proteins interactions, the structures do not possess functionality which would be sufficient to direct binding to STAT5 selectively. It would seem likely that they might bind to many other enzymatic active sites. This should be discounted or selectively demonstrated on a wider range of protein substrates.

In order to study the selectivity of the novel inhibitors for STAT5, we have first tested a list of closely related, homologous proteins (STAT3, STAT1), other proteins with phosphotyrosine binding sites (SHP2, PTP1B), and for control the used protein tag (MBP). In none of these proteins we found binding or inhibition.

In the next step, we tested the binding of inhibitors in cellular lysates using both CETSA and photo-crosslinking experiments (for details see answer to point 1).

Finally, the selectivity of the STAT5 inhibitors was investigated in living cells using the isothermal dose response finger prints (ITDRF) (Figure 5f,g). The cellular occupancy (OC50) of STAT5 protein in the cells was close to the observed IC50 value of cellular proliferation, indicating that indeed binding of the inhibitors to STAT5 – and not to other targets - was responsible for the observed phenotype. In addition, the results of the siRNA experiments suggested that the effect of compound **16** on cell viability was exerted by inhibition of STAT5 and not by additional off-target effects. Furthermore, compound **16** affected cell viability only in cell lines known to be driven by STAT5, i.e. K562 (human CML cell line), MV-4-11 (human AML cell line), and BaF3/FLT3-ITD (SFigure 6a-f) while it did not exhibit cytotoxicity toward normal epithelial cells and other types of cancers that are not driven by STAT5 (SFigure 6g-k).

5. The fragments contain metabolically labile groups (prior/after the Mannich reaction), such as a carboxylate for instance. The authors should consider exploring the clearance rates of these fragments in an in vitro hepatocyte assay.

The carboxylate group is considered by medicinal chemists as a group that is well-acceptable in drugs and in fact is part of numerous drugs which are used in billions of prescribed daily doses or as OTC-drugs every year. Examples include most ACE inhibitors used these days (like Enalapril, Ramipril, etc.), other highly prescribed antihypertensives like Valsartan, the major antihistaminic Cetirizine or also acetyl-salicylic acid and Ibuprofen. For our inhibitors, we find as the major route of clearance the acid-catalyzed hydrolysis of the N,N-aminal structure at strongly acidic conditions at pH-values <3.0.

Minor Comments

- Authors used a dose of 200 mg/kg (no toxicity was observed in mouse study), however this does seem as quite a large dose, a comment on the reason behind this? Or scalability in the future?

We have used this high concentration due to the good solubility and low toxicity of our compounds in order to maximize the effect of our compounds in-vivo in this proof-of-concept study. In the future this dose might be reducible.

- The Western blots presented in Figure 4 (especially, f. IV) are hard to visualize and poor quality. The bands are almost unrecognizable and inconclusive (may require a repeat).

We have repeated these experiments and now present the data in a much clearer SDS PAGE and Western blot.

Reviewer #3 (Remarks to the Author):

The manuscript by Rademann and colleagues reports an implementation of the protein-templated ligation in order to discover new inhibitors for STAT5 transcription factor. Such transcription factor being involved in various cancers is considered as a difficult drug target. Remarkably, in the manuscript authors report the discovery of a series of compounds that can bind to this protein and inhibit its phosphorylation, thereby affecting its function. The study is performed rigorously and the findings are of high significance. The implementation of three-component Mannich reaction to produce libraries of inhibitors is particularly innovative. Overall, the manuscript can be recommended for publication with minor modifications.

Thank you for this generally positive evaluation of our work.

Comments:

1) On page 5, lines 101-107 the description of in situ assays for Mannich ligation is confusing. First, three components for ligation are added at pH 5, then protein MBP-STAT5b is added in buffer at pH 7.4. What is the actual pH value of the catalyzed ligation in the assays? Authors stated that background reaction (without protein) proceeds spontaneously at pH 5 and does not occur at pH 7.4, while upon addition of STAT5b reaction can occur. The current description gives an impression that pH is between 5 and 7, which may result in significant background reaction.

The in-situ reactions were carried out under slightly acidic conditions at a pH of 5 obtained by mixing fragment **3** with FA and tetrazole in water. After addition of 50 mM MOPS buffer to the in-situ reactions, the pH was stable at 7.4. No background reaction was detected at pH 7.4 unless STAT5 protein was present. We have slightly revised the description of these experiments in order to clarify the experimental conditions (page 5 of the manuscript, lines 104-108).

2) Page5, line 108, "Figure 2b": in the actual Figure 2 there is no "b", the correct designation should be added; "SData" should be "SData1" or "SData2"?

We have added the correct designations in Figure 2.

In summary, we would like to thank all reviewers for their valuable input, which has enabled us to further improve the quality and the conclusiveness of our manuscript. We hope that we have answered and clarified all questions and comments to your full satisfaction and are looking forward to your reply.

With best regards,

Jörg Rademann

REVIEWERS' COMMENTS:

Reviewer #1 (Remarks to the Author):

The authors have satisfactorily addressed critiques and the manuscript is now acceptable for publication.

Reviewer #2 (Remarks to the Author):

The authors have addressed all the major concerns. Most importantly they prepared the mutant protein and conducted comparative experiments which support their hypothesis.

I believe the manuscript is now suitable for publication.

Reviewer #3 (Remarks to the Author):

The manuscript has been carefully revised. All the points requested by the reviewers are addressed. The manuscript can be recommended for publication.